# Neurological Symptoms That May Represent a Warning in Terms of Diagnosis and Treatment in a Group of Children and Adolescents with Vitamin D Deficiency

**DOI:** 10.3390/children10071251

**Published:** 2023-07-20

**Authors:** Oguzhan Korkut, Hilal Aydin

**Affiliations:** 1Department of Medical Pharmacology, Faculty of Medicine, Balikesir University, 10145 Balikesir, Turkey; 2Department of Pediatrics, Faculty of Medicine, Balikesir University, 10145 Balikesir, Turkey; hilal.aydin@balikesir.edu.tr

**Keywords:** vitamin D, vitamin D deficiency, neurology, pediatrics, pharmacology

## Abstract

Aim: This research was intended to evaluate the clinical and laboratory findings of children presenting to our pediatric neurology clinic with symptoms potentially linked to vitamin D deficiency and with low vitamin D levels and the distribution of those findings by sex, age groups, and vitamin D levels. Methods: This retrospective study involved patients presenting to our clinic with symptoms potentially associated with vitamin D deficiency and low serum concentrations of 25 OH vitamin D (25 OH D) (<75 nmol/L, 30 µg/mL). Patients’ movement disorders and central nervous system-related symptoms at the time of presentation and serum 25 OH D, calcium (Ca), phosphorus (P), and magnesium (Mg) levels were recorded and evaluated in terms of age, sex, and vitamin D levels. Results: Eight hundred twenty-two cases of vitamin D deficiency were included in the study, 50.2% (*n* = 413) boys and 49.8% (*n* = 409) girls. Although cases of vitamin D deficiency were present across all the age groups between 1 and 18, they were most common in the 5–14 age range (*n* = 372, 45.3%). Movement disorders were observed in 14.6% (*n* = 120) of our cases, and neurological findings associated with the central nervous system were observed in 52.6% (*n* = 432). The most common accompanying movement in our cases was difficulty remaining in balance (*n* = 42, 35%), while the most frequent accompanying central nervous system finding was vertigo (*n* = 99, 22.92%). Other movement disorders encountered included limb shaking (*n* = 32, 26.7%), abnormal posture (*n* = 20, 16.67%), easy falling (*n* = 16, 13.33%), body rigidity (*n* = 15, 12.5%), and hand clenching (*n* = 5, 4.17%). Other frequently encountered neurological findings were headache (*n* = 88, 20.37%), epileptic seizures (*n* = 83, 19.21%), fainting (*n* = 58, 13.43%), developmental delay (*n* = 41, 9.49%), febrile seizures (*n* = 33, 7.64%), and numbness in the fingers (*n* = 20, 4.63%). Other neurological findings were sleep disorders (*n* = 10, 2.31%), nightmares (*n* = 8, 1.85%), pain in the extremities (*n* = 7, 1.62%), and sweating and frailty (*n* = 4, 0.93% for both). Ca, P, and Mg levels were lower in cases with vitamin D levels < 12 µg/mL. The prevalences of both movement disorders and central nervous system findings varied according to age groups, sex, and vitamin D levels. Conclusions: Our study results show that vitamin D deficiency can present with different neurological findings and that these may vary according to age group, sex, and vitamin D levels. Clinicians must take particular care in pediatric cases with neurological findings in terms of the early diagnosis and treatment of vitamin D deficiency.

## 1. Introduction

Vitamin D is a fat-soluble substance with a major function in regulating calcium and phosphorus homeostasis and protecting bone health. Deficiency is a significant global health problem reported in 25–50% of the general population [1]. Due to the rapid rate of growth and development involved, childhood vitamin D requirements are higher than those of adults, and pregnant women and the elderly are both in the high-risk group in terms of vitamin D deficiency [2]. In addition to increasing requirements, vitamin D deficiency is closely linked to nutrition, vitamin absorption, factors affecting metabolism, and the degree of exposure of the skin to sunlight. The prevalence of vitamin D deficiency varies depending on age, nutritional habits, malabsorption syndromes, and even geographical regions. For example, the reported prevalence of vitamin deficiency is 85–98% among adolescents in India, 96% in Saudi Arabia, and 62.2% in a study from Turkey [3,4,5].

Vitamin D deficiency and the treatment thereof have been the subject of intense research for many years. While vitamin D has long been known to be involved in the regulation of calcium and phosphorus homeostasis and in bone health, it has also been found to play a role in various system functions, such as immunity, and in several pathophysiological processes. The potential link between vitamin D deficiency and various diseases is still being investigated and discussed [6,7]. Vitamin D deficiency generally leads to rickets in children and osteomalacia in adults, and some observational studies have also shown an association with various diseases such as asthma and allergy, diabetes, depression, and cancer, and even with mortality [8,9,10]. Randomized controlled studies have also shown that, while the evidence indicating that vitamin D supplementation reduces the risk of cancer, depression, asthma, diabetes, cardiovascular disease, and general mortality is unclear, resolving the deficiency improves the clinical findings of several diseases [9,10,11].

Less is understood concerning the potential effects of vitamin D deficiency on the central nervous system and its functions compared to other systems. However, studies have shown that deficiency may be linked to neurological function disorders [12]. Research has reported that children with severe vitamin D deficiency and rickets may present with symptoms of motor development delay, muscle hypotonia, and weakness [13,14], and growth and development retardation, lethargy, and hypocalcemic seizures can be seen in these children [15]. Studies have also reported higher rates of vitamin D deficiency in individuals with neurodevelopmental disorders and mental disability [16].

Adequate vitamin D intake and the maintenance of optimal levels are currently thought to be important, not solely for calcium and phosphorus homeostasis and bone health but also for the preservation of general health, including numerous systems. Deficiency is, therefore, very important and a situation requiring prompt detection and resolution. However, since unnecessary tests pose a burden on national health systems and can result in loss of working productivity, and due to the risk of toxicity posed by unnecessary vitamin D supplementation, patients’ clinical status should be evaluated in terms of vitamin D deficiency. It should also not be forgotten that patients with vitamin D deficiency may present with symptoms and clinical findings involving systems other than the musculoskeletal system.

The number of studies examining the relationship between vitamin D deficiency and neurological findings in children is very low. The present study was conducted in order to evaluate the clinical and laboratory findings of children presenting to the newly established pediatric neurology clinic in our hospital with complaints potentially associated with vitamin D deficiency and with low vitamin D levels and to examine the distributions of those findings in terms of sex, age groups, and vitamin D levels.

## 2. Materials and Methods

### 2.1. Study Design

Our study was performed retrospectively with children and adolescent patients presenting to the Balıkesir University Medical Faculty Child Health and Diseases Neurology Clinic, Turkey, between 1 August 2019 and 1 September 2022.

Patients with central nervous system findings and/or movement disorders and with low serum 25 OH vitamin D concentrations (<75 nmol/L [30 µg/mL]) were included. Patients with chronic diseases capable of affecting vitamin D levels, regularly using drugs, or with missing file data were excluded.

Patients’ complaints concerning movement disorders and the central nervous system at the time of admission and their serum 25 OH vitamin D (25 OH D), calcium (Ca), phosphorus (P), and magnesium (Mg) levels as laboratory parameters were recorded. The patients’ clinical and laboratory findings were examined in terms of age groups, sex, and vitamin D levels.

Ethical committee approval was received prior to the commencement of the study (no: 2022/97).

### 2.2. Statistical Analysis

The study data were analyzed on SPSS version 17.0 software (SPSS Inc., Chicago, IL, USA). The variables were calculated as number (*n*), percentage (%), and mean ± standard deviation (SD). The chi-square, Mann–Whitney, and Kruskal–Wallis tests were applied to identify any differences in clinical findings and laboratory parameters between the sexes, age groups, and vitamin D levels. *p* values < 0.05 were considered statistically significant.

## 3. Results

Eight hundred twenty-two patients presenting to the Balıkesir University Medical Faculty Child Health and Diseases neurology clinic with symptoms potentially associated with vitamin D deficiency and with low serum 25 OH vitamin D concentrations were enrolled in the study. Boys constituted 50.2% (*n* = 413) of these patients, and girls 49.8% (*n* = 409). The mean age of the cases was 11 ± 5.20 (1–18 years). Patients were most frequently in the 5–14 age group (*n* = 372, 45.3%), followed by the 14–18 age group (*n* = 273, 33.2%) and the 2–5 age group (*n* = 147, 14.7%). The group with the fewest patients was the 0–2 age range (*n* = 30, 3.6%). Girls were most frequently in the 14–18 age range (*n* = 176, 43%) and boys in the 5–14 age range (*n* = 210, 50.8%). The lowest number of patients in both sexes was in the 0–2 age range (Table 1).

Examination of our patients’ laboratory results showed that the mean serum 25 (OH) D concentration among all the patients was 18.36 ± 6.26 (3.04–30) µg/mL, while the mean Ca, P, and Mg levels were 9.83 ± 0.40, 4.54 ± 0.76, and 2.07 ± 0.27 mg/dL, respectively (Table 1).

Mean 25 OH D concentrations were 19.46 ± 6.08 (3.17–30) µg/mL in boys and 17.25 ± 6.26 (3.04–30) µg/mL in girls. Mean Ca, P, and Mg levels were 9.85 ± 0.39, 4.67 ± 0.75, and 2.09 ± 0.29 mg/dL, respectively, in boys and 9.80 ± 0.40, 4.41 ± 0.75, and 2.05 ± 0.23 mg/dL in girls. Girls exhibited significantly lower 25 OH D, P, and Mg levels than boys (*p* = 0.001, *p* = 0.001, and *p* = 0.004, respectively) (Table 1).

The patients were assigned into one of three groups based on their vitamin D levels—those with 25 (OH) levels <12 µg/mL, 12–20 µg/mL, and >20–30 µg/mL. Our patients’ 25 (OH) D levels were most frequently in the 12–20 µg/mL range (*n* = 348, 42.3%), followed by the >20–30 µg/mL range (*n* = 329, 40%), while the lowest number of patients was in the <12 µg/mL 25 (OH) D range (*n* = 145, 17.6%). Additionally, 25 (OH) D levels in girls were most frequently in the 12–20 µg/mL range (*n* = 182, 44.5%), while in boys, they were most frequently in the 20–30 µg/mL (*n* = 195, 47.2%) range. Fewer patients were in the <12 µg/mL ranges in both sexes (22.7% in girls and 12.6% in boys) (Table 1).

Various types of movement disorders were present in 120 (14.6%) of our cases. More than one such disorder was observed in some patients. The most common accompanying movement disorder was difficulty remaining in balance (*n* = 42, 35%), followed by limb shaking (*n* = 32, 26.7%). Movement disorders were observed in 17.7% (*n* = 73) of boys and 11.5% (*n* = 47) of girls (*p* = 0.024). The most common movement disorder in boys was difficulty remaining in balance (*n* = 28, 38.36%), while the most frequent disorders in girls were limb shaking (*n* = 17, 36.17%) and difficulty remaining in balance (*n* = 14, 29.79%) (Table 2). Accompanying central nervous system neurological findings were present in 52.6% (*n* = 432) of our cases, with more than one such finding being observed in some. The most common accompanying neurological findings, in descending order, were vertigo (*n* = 99, 22.92%), headache (*n* = 88, 20.37%), and epileptic seizures (*n* = 83, 19.21%). Neurological findings were seen in 44.3% of boys but at a significantly higher rate in girls (60.9%) (*p* = 0.001). Epileptic seizures were the most frequent accompanying finding in boys (*n* = 44, 24.04%), and vertigo (*n* = 67, 26.91%) and headache (*n* = 63, 25.30%) in girls (Table 2).

When our patients were classified on the basis of their vitamin D levels, individuals with values < 12 µg/mL were most frequently in the 14–18 age group (*n* = 68, 46.9%), followed by the 5–14 age group (*n* = 66, 45.5%) (Table 3). Movement disorder was present in only 15.2% of our patients with vitamin D levels < 12 µg/mL, while accompanying central nervous system neurological findings were present in 62.1%. The most common movement disorder in this group was difficulty remaining in balance (*n* = 10, 45.45%), while the most common central nervous system finding was vertigo (*n* = 26, 28.89%) (Table 4). Our cases with vitamin D levels of 12–20 µg/mL and >20–30 µg/mL were most frequently in the 5–14 age range (46.6% and 43.8, respectively). The incidence of movement disorder in these vitamin D levels was 14.1% and 14.9%, respectively. Accompanying central nervous system neurological findings were present in 54% and 46.8% of these patients. The most frequent movement disorders at vitamin D levels of 12–20 µg/mL were difficulty remaining in balance (*n* = 19, 38.78%) and limb shaking (*n* = 13, 26.53%), while in individuals with vitamin D levels of >20–30 µg/mL, the most frequent disorders were abnormal posture (*n* = 13, 26.53%), difficulty remaining in balance (*n* = 13, 26.53%), and limb shaking (*n* = 11, 22.45%). The most frequent accompanying neurological findings in our patients at both vitamin D levels were vertigo, headache, and epileptic seizures. No difference in terms of the incidence of movement disorders was observed in the vitamin D groups (*p* = 0.934), but the incidence of central nervous system neurological findings varied significantly (*p* = 0.007) (Table 3 and Table 4). In terms of laboratory parameters, significant variation was observed in Ca and P levels between the vitamin D groups (*p* = 0.00 and *p* = 0.002, respectively). Ca and P levels were lowest in the patients with vitamin D values < 12 µg/mL. Mg levels were also lower in this group, although this was not statistically significant (Table 3).

The analysis also revealed no significant difference in the incidence of movement disorders in terms of age groups, although central nervous system neurological findings differed significantly (*p* = 0.597 and *p* = 0.010, respectively). The incidences of movement disorders in the 0–2, >2–5, >5–14, and >14–18 age groups were 10%, 18.4%, 14.8%, and 12.8, respectively, and the incidences of neurological findings were 40%, 47.6%, 49.7%, and 60.4%. The most frequent movement disorders were difficulty remaining in balance and limb shaking, and the most frequent central nervous system neurological findings were vertigo and epileptic seizures. Neurological findings were most common in the >14–18 age range (60.4%). Mean 25 (OH) D, Ca, P, and Mg levels also differed significantly in the age groups (*p* = 0.001) (Table 5 and Table 6).

## 4. Discussion

Vitamin D is a steroid-type vitamin with significant involvement in the regulation of calcium and phosphorus homeostasıs [1]. While it can be absorbed from plant (ergocalciferol, D2) and animal (cholecalciferol, D3) products, it is essentially synthesized in the skin from 7-dehydrocholesterol [17]. Oxidation, reduction reactions, and cytochrome P450 (CYP450) enzymes perform functions in the metabolism and synthesis of vitamin D. While 25 (OH) D derivatives result from hydroxylation in the liver, 1.25 (OH2) D derivatives representing active vitamin D are formed as a result of secondary hydroxylation with 1 alpha hydroxylase enzyme in the kidney, placenta, immune cells, and brain [18]. Vitamin D exhibits its metabolic effects by means of intracellular vitamin D receptors (VDRs) in numerous tissues [1].

While the importance of vitamin D in calcium and phosphorus homeostasis and the musculoskeletal system are well known, the presence of VDRs has also been shown in several tissues such as the pituitary gland, thyroid, breast, heart muscle, liver, kidney, skin, colon and small intestine, gonads, osteoblasts, mononuclear cells, lymphocytes, and pancreatic islet cells in recent years; it has also been shown play an important role in systems such as the gastrointestinal, cardiovascular, endocrine, and immune systems and in numerous pathophysiological processes [19].

Although the effects of vitamin D on the central nervous system and neurological functions are so far little understood, studies have demonstrated its involvement in cerebral morphology and function in both embryonic and adult life and confirmed that it is essential for brain development [20]. VDRs are widely present in the cortex, amygdala, thalamus, and hippocampus, and 1-α-hydroxylase is expressed in the substantia nigra [20]. The extensive expression of VDRs in both fetal and adult brain tissues supports the importance of vitamin D in brain functions [21]. Indeed, studies have shown an association between vitamin D deficiency and neurological function compromise. In the presence of vitamin D deficiency, patients may develop neuromuscular sensitivity such as fatigue and muscle pain, numbness, and even contractions in the extremities, depending on the degree of hypocalcemia involved [22,23].

Research has shown that maternal vitamin D deficiency impairs fetal brain development in animal experiments, and structural anomalies such as decreased growth in the lateral ventricle and decreased cortical thickness have been observed in offspring [24,25]. Learning and recall processes can also be adversely affected [24]. Studies have similarly confirmed the involvement of vitamin D in neural communication, which is associated with reward-related locomotor and emotional behavior and cognition [25,26]. VDR expression also occurs in areas of the brain that regulate the sleep/wakefulness cycle, such as the hypothalamus, prefrontal cortex, and substantia nigra, and epidemiological studies have shown that vitamin D deficiency can lead to sleep disorders, while vitamin D supplementation prevents and can ameliorate such disorders [27]. In addition to its effects on brain development and functions, vitamin D also exhibits neuroprotective effects through direct and indirect mechanisms. Vitamin D exhibits anti-inflammatory effects by suppressing the release of proinflammatory cytokines by activated microglia in the brain, inhibiting nitric oxide (NO) synthesis during neurodegenerative diseases and ischemia, and also reducing the oxidative load in neurons and microglia by increasing γ-glutamyl-transpeptidase [28,29]. The presence of low serum vitamin D levels in neurodegenerative, neuroinflammatory, and neuropsychological diseases also corroborates the neuroprotective effects of vitamin D and its significant role in brain functions, suggesting potential involvement in the pathogenesis of these diseases [30,31]. Studies have even reported that some headaches have been reduced by means of vitamin D supplementation and that these headaches may be associated with inflammation and neuromuscular sensitivity linked to vitamin D deficiency [32]. In addition to all these well-known effects, which are still under investigation, studies suggest that 25–50% of the general population suffers from vitamin D deficiency, representing a significant global public health problem [1]. Studies in different age groups have also shown that high rates of vitamin D insufficiency and deficiency are encountered in pediatric patients [2]. Serum 25 (OH) D levels are measured in the current evaluation of vitamin D sufficiency since these indicate both vitamin D intake and its endogenous production and half-life (2–3 weeks). A serum 25 (OH) D concentration <25 nmol/L (12 ng/mL) in patients is regarded as indicating severe vitamin D deficiency, concentrations of 25–50 nmol/L (12–20 ng/mL) moderate deficiency, and concentrations of 50–75 nmol/L (20–30 ng/mL) insufficiency [33].

The present study investigated the neurological findings with which patients presented, their laboratory findings and the age groups concerned with those findings and their distributions according to sex and vitamin D levels in patients presenting to our clinic with symptoms potentially associated with vitamin D deficiency and with low serum 25 (OH) D concentrations. The study involved 822 patients aged up to 18 and with vitamin D deficiency. Boys represented 50.2% of the cases, and girls 49.8% (Table 1).

Studies have shown that vitamin D deficiency may be encountered in all pediatric age groups. However, Basu et al. and Al-Taiar et al. reported that 25 (OH) D levels decreased with age, the lowest levels being observed in adolescence [34,35]. In a study from China, Zhu et al. reported that vitamin D deficiency was most frequently observed in the 6–16 age range [36]. Consistent with these studies, the cases of vitamin D deficiency in the present research were most commonly in the 5–14 age range (*n* = 372, 45.3%). The lowest number of cases was in the 0–2 age group (*n* = 30, 3.6%), while 147 cases were in the 2–5 age group (14.7%) and 273 in the 14–18 age group (33.2%) (Table 1).

Basu et al. determined no significant difference between male and female pediatric cases in terms of serum vitamin D levels [34], while in the present study, and in agreement with Al-Taiar et al. [35], Neyestani et al. [37] and Isa et al. [38], girls exhibited lower serum 25 (OH) D levels than boys (17.24 ± 6.24 and 19.46 ± 6.08 µg/mL, respectively, *p* = 0.001) (Table 1).

Neyestani et al. determined no statistically significant difference in levels of Ca or P between boys and girls with vitamin D deficiency, although Mg levels were significantly lower in boys [37]. In the present study, Ca, P, and Mg levels were all lower in girls than in boys, the difference in P and Mg levels being statistically significant (*p* = 0.001 and *p* = 0.004, respectively) (Table 1). Zhang et al. reported movement disorders in 37.7% of the cases in their study and accompanying central nervous system findings in 62.3% [39]. Movement disorders were present in 14.6% of the cases in the present study, and central nervous system findings in 52.6%. Zhang et al. described limb shaking as the most common movement disorder and headache as the most frequent central nervous system finding [39]. In the present study, the most frequent movement disorder was difficulty in remaining in balance (*n* = 42, 35%), and the most common central nervous system finding was vertigo (*n* = 99, 22.92%). The second most frequent movement disorder was limb shaking (*n* = 32, 26.7%), and other frequent accompanying neurological findings were headache (*n* = 88, 20.37%), epileptic seizures (*n* = 83, 19.21%), fainting (*n* = 58, 13.43%), developmental delay (*n* = 41, 9.49%), febrile seizures (*n* = 33, 7.64%), and finger numbness (*n* = 20, 4.63%) (Table 2).

The incidences of movement disorders and neurological findings differed significantly according to sex in the present study (*p* = 0.024 and *p* = 0.000, respectively). Accompanying central nervous system findings were present in 44.3% of boys but in 60.9% of girls. Movement disorders were more frequent in boys (17.7%). The most frequent accompanying neurological findings in boys were epileptic seizures (*n* = 44, 24.04%) followed by vertigo (*n* = 32, 17.49%), while in girls, they were vertigo (*n* = 67, 26.91%) followed by headache (*n* = 63, 25.30%). The most frequent movements were difficulty remaining in balance in boys (*n* = 28, 38.36%) and limb shaking in girls (*n* = 17, 36.17%) (Table 2). The age groups in this study exhibited no significant differences in terms of the incidence of movement disorders; the incidence of central nervous system findings varied significantly (*p* = 0.597 and *p* = 0.010, respectively). The incidences of movement disorders in the 0–2, >2–5, >5–14, and >14–18 age groups were 10%, 18.4%, 14.8%, and 12.8%, respectively, while the figures for neurological findings were 40%, 47.6%, 49.7%, and 60.4%. The most frequent neurological findings were headache in the 14–18 age group (*n* = 36, 21.82%), vertigo in the 8–14 age group (*n* = 50, 27.03%), and epileptic seizures in the 0–2 and 2–5 age groups (66.67% and 25.71%, respectively) (Table 6).

### Limitations of the Study

A number of limitations should be considered when interpreting this study. These include the fact that it was performed in a single center and that various factors capable of affecting vitamin D levels were not investigated (such as PTH levels, liver and kidney function tests, children’s BMI, seasonal factors related to exposure to sunshine, living in rural or urban areas, diet, use of nutritional support and physical activity. Further studies involving more centers, with larger patient numbers, and including factors capable of impacting vitamin D levels are now required to clarify the neurological presentations of vitamin D deficiency.

## 5. Conclusions

In conclusion, vitamin D deficiency is widespread in childhood. While unnecessary vitamin supplementation entails a risk of toxicity, unnecessary 25(OH) D level measurements can impose a burden on health systems and cause work productivity losses. From that perspective, it is important for clinicians to know that vitamin D deficiency can present with different clinical characteristics and to pay careful attention to early diagnosis and treatment. From the neurological perspective, it is recommended that evaluations in terms of vitamin D deficiency be performed in cases of locomotor system disorders, developmental delay, cerebral palsy, chronic immobilization, autism, and disorders such as multiple sclerosis, seizures of unknown etiology, epilepsy, myopathy, and muscular dystrophy [40,41]. Such evaluation is necessary since, in addition to vitamin D deficiency being capable of playing a role in the pathophysiology of diseases, levels can also change in association with various drugs employed in treatment. However, it should also be remembered that deficiency can also emerge with various clinical findings, even in the absence of chronic disease and drug use.

Movement disorders were observed in 14.6% (*n* = 120) of our cases, with vitamin D deficiency and neurological findings associated with the central nervous system in 52.6% (*n* = 432). The incidences of these findings varied depending on the patients’ age group, sex, and vitamin D levels. The fact that headaches, the incidence of which was determined in this study, can be treated with vitamin D supplementation [32,42] shows the importance of headaches and other clinical presentations in both the diagnosis of deficiency and early treatment. The number of studies in the literature examining neurological presentations that may develop in association with vitamin D deficiency is quite small, and the numbers of cases are low. From that perspective, we think that this study will make an important contribution to the literature.

## Figures and Tables

**Table 1 children-10-01251-t001:** Age distributions and laboratory parameters by sex.

	All Patients (*n* = 822)	Girls (*n* = 409, 49.8%)	Boys (*n* = 413, 50.2%)	*p*
Mean age	11 ± 5.20	12.25 ± 5.09	9.76 ± 5.02	
0–2 years (*n*, %)	30 (3.6%)	13 (3.2%)	17 (4.1%)	
>2–5 years (*n*, %)	147 (17.9)	58 (14.2%)	89 (21.5%)	
>5–14 years (*n*, %)	372 (45.3%)	162 (39.6%)	210 (50.8%)	
>14–18 years (*n*, %)	273 (33.2%)	176 (43%)	97 (23.5%)	
25 OH D				
<12 µg/mL (*n*, %)	145 (17.6%)	93 (22.7%)	52 (12.6%)	
12–20 µg/mL (*n*, %)	348 (42.3%)	182 (44.5%)	166 (40.2%)	
>20–30 µg/mL (*n*, %)	329 (40%)	134 (32.8%)	195 (47.2%)	
Mean 25 OH D (µg/mL)	18.36 ± 6.26 (3.04–30)	17.25 ± 6.26 (3.04–30)	19.46 ± 6.08 (3.17–30)	0.001 ***
Mean Ca (mg/dL)	9.83 ± 0.40 (8.4–11)	9.80 ± 0.40 (8.5–10.8)	9.85 ± 0.39 (8.4–11)	0.090
Mean P (mg/dL)	4.54 ± 0.76 (2–6.95)	4.41 ± 0.75 (2.5–6.6)	4.67 ± 0.75 (2–6.95)	0.001 ***
Mean Mg (mg/dL)	2.07 ± 0.27 (1.8–6)	2.05 ± 0.23 (1.8–5)	2.09 ± 0.29 (1.8–6)	0.001 ***

Note: 25 OH D: 25 OH vitamin D; Ca: Calcium; P: Phosphorus; Mg: Magnesium; *** *p* = 0.001.

**Table 2 children-10-01251-t002:** The frequencies of movement disorders and central nervous system findings by gender.

	All Patients (*n* = 822)	Girls (*n* = 409, 49.8%)	Boys (*n* = 413, 50.2%)	*p*
Movement Disorders				0.024 *
No	702 (84.9)	362 (88.5%)	340 (82.3%)	
Yes	120 (14.6)	47 (11.5%)	73 (17.7%)	
Limb shaking	32 (26.7%)	17 (36.17%)	15 (20.55%)	
Hand clenching	5 (4.17%)	2 (4.26%)	3 (4.11%)	
Body rigidity	15 (12.5%)	7 (14.89%)	8 (10.96%)	
Difficulty remaining in balance	42 (35%)	14 (29.79%)	28 (38.36%)	
Abnormal posture	20 (16.67%)	8 (17.02%)	12 (16.44%)	
Easy falling	16 (13.33%)	6 (12.77%)	10 (13.70%)	
Central nervous system findings				0.001 ***
No	390 (47.4%)	160 (39.1%)	230 (55.7%)	
Yes	432 (52.6%)	249 (60.9%)	183 (44.3%)	
Pain in extremities	7 (1.62%)	2 (0.80%)	5 (2.73%)	
Vertigo	99 (22.92%)	67 (26.91%)	32 (17.49%)	
Finger numbness	20 (4.63%)	13 (5.22%)	7 (3.83%)	
Sweating	4 (0.93%)	2 (0.80%)	2 (1.09%)	
Febrile seizures	33 (7.64%)	15 (6.02%)	18 (9.84%)	
Fainting	58 (13.43%)	41 (16.47%)	17 (9.29%)	
Epileptic seizures	83 (19.21%)	39 (15.66%)	44 (24.04%)	
Nightmares	8 (1.85%)	6 (2.41%)	2 (1.09%)	
Developmental delay	41 (9.49%)	11 (4.42%)	30 (16.39%)	
Frailty	4 (0.93%)	2 (0.80%)	2 (1.09%)	
Headache	88 (20.37%)	63 (25.30%)	25 (13.66%)	
Sleep disorder	10 (2.31%)	4 (1.61%)	6 (3.28%)	

* *p* < 0.05; *** *p* = 0.001.

**Table 3 children-10-01251-t003:** Age, sex, and laboratory parameters according to vitamin D levels.

	25 (OH) D <12 (µg/mL) (*n* = 145, 17.6%)	25 (OH) D 12–20 (µg/mL) (*n* = 348, 42.3%)	25 (OH) D 20–30 (µg/mL) (*n* = 329, 40%)	*p*
Age groups				
0–2 (*n*, %)	2 (1.4%)	11 (36.7%)	17 (5.2%)	
>2–5 (*n*, %)	9 (6.2%)	54 (15.5%)	84 (25.5%)	
>5–14 (*n*, %)	66 (45.5%)	162 (46.6%)	144 (43.8%)	
>14–18 (*n*, %)	68 (46.9%)	121 (44.3%)	84 (25.5%)	
Sex				
Female (*n*, %)	93 (22.7%)	182 (44.5%)	134 (32.8%)	
Male (*n*, %)	52 (12.6%)	166 (40.2%)	195 (47.2%)	
Mean 25 (OH) D (µg/mL)	9.31 ± 2.15	16.14 ± 2.29	24.7 ± 2.94	0.001 ***
Mean Ca (mg/dL)	9.65 ± 0.43	9.82 ± 0.36	9.91 ± 0.39	0.001 ***
Mean P (mg/dL)	4.34 ± 0.67	4.56 ± 0.76	4.61 ± 0.79	0.002 **
Mean Mg (mg/dL)	2.04 ± 0.16	2.08 ± 0.25	2.07 ± 0.31	0.636

Note: 25 OH D: 25 OH vitamin D; Ca: Calcium; P: Phosphorus; Mg: Magnesium; ** *p* < 0.01; *** *p* = 0.001.

**Table 4 children-10-01251-t004:** Frequencies of movement disorders and central nervous system findings according to vitamin D levels.

	25 (OH) D <12 (µg/mL) (*n* = 145, 17.6%)	25 (OH) D 12–20 (µg/mL) (*n* = 348, 42.3%)	25 (OH) D 20–30 (µg/mL) (*n* = 329, 40%)	*p*
Movement disorder				0.934
No	123 (84.8%)	299 (85.9%)	279 (84.8%)	
Yes	22 (15.2%)	49 (14.1%)	49 (14.9%)	
Limb shaking	8 (36.36%)	13 (26.53%)	11 (22.45%)	
Hand clenching	1 (4.55%)	2 (4.08%)	2 (4.08%)	
Body rigidity	3 (13.64%)	7 (14.29%)	5 (10.20%)	
Difficulty remaining in balance	10 (45.45%)	19 (38.78%)	13 (26.53%)	
Abnormal posture	4 (18.18%)	3 (6.12%)	13 (26.53%)	
Easy falling	3 (13.64%)	6 (12.24%)	7 (14.29%)	
Central nervous system findings				0.007 **
No	55 (37.9%)	160 (46.0%)	175 (53.2%)	
Yes	90 (62.1%)	188 (54.0%)	154 (46.8%)	
Pain in extremities	2 (2.22%)	4 (2.13%)	1 (0.65%)	
Vertigo	26 (28.89%)	46 (24.47%)	27 (17.53%)	
Numbness in fingers	6 (6.67%)	8 (4.26%)	6 (3.90%)	
Sweating	1 (1.11%)	3 (1.60%)	-	
Febrile seizures	6 (6.67%)	7 (3.72%)	20 (12.99%)	
Fainting	16 (17.78%)	24 (12.77%)	18 (11.69%)	
Epileptic seizures	14 (15.56%)	36 (19.15%)	33 (21.43%)	
Nightmares	2 (2.22%)	4 (2.13%)	2 (1.30%)	
Developmental delay	6 (6.67%)	14 (7.45%)	21 (13.64%)	
Frailty	3 (3.33%)	1 (0.53%)	-	
Headache	15 (16.67%)	45 (23.94%)	28 (18.18%)	
Sleep disorder	2 (2.22%)	2 (1.06%)	6 (3.90%)	

** *p* < 0.01.

**Table 5 children-10-01251-t005:** Sex and laboratory parameters by age groups.

	0–2 Age	>2–5 Age	>5–14 Age	>14–18 Age	*p*
Gender					
Female (*n*, %)	17 (56.7%)	58 (39.5%)	162 (43.5%)	176 (64.5%)	
Male (*n*, %)	13 (43.3%)	89 (60.5%)	210 (56.5%)	97 (35.5%)	
Mean 25 (OH) D (µg/mL)	21.32 ± 6.34	21.15 ± 5.80	18.31 ± 6.17	16.59 ± 5.98	0.001 ***
Mean Ca (mg/dL)	10.17 ± 0.36	9.90 ± 0.41	9.81 ± 0.40	9.78 ± 0.37	0.001 ***
Mean P (mg/dL)	5.53 ± 0.82	5.13 ± 0.54	4.70 ± 0.59	3.92 ± 0.58	0.001 ***
Mean Mg (mg/dL)	2.12 ± 0.16	2.16 ± 0.42	2.05 ± 0.13	2.05 ± 0.29	0.001 ***

*** *p* = 0.001.

**Table 6 children-10-01251-t006:** The frequency of movement disorders and central nervous system findings by age groups.

	0–2 Age *n* (%)	>2–5 Age *n* (%)	>5–14 Age *n* (%)	>14–18 Age *n* (%)	*p*
Movement disorder					0.597
No	27 (90%)	120 (81.6)	317 (85.2%)	238 (87.2%)	
Yes	3 (10%)	27 (18.4)	55 (14.8)	12.8%)	
Limb shaking	2 (66.67%)	6 (22.22%)	12 (21.82%)	12 (34.29%)	
Hand clenching	-	2 (7.41%)	3 (5.45%)	-	
Body rigidity	-	3 (11.11%)	6 (10.91%)	6 (17.14%)	
Difficulty remaining in balance	-	9 (33.33%)	21 (38.18%)	12 (34.29%)	
Abnormal posture	-	5 (18.52%)	10 (18.18%)	5 (14.29%)	
Easy falling	1 (33.33)	7 (25.93%)	6 (10.91%)	2 (5.71%)	
Central nervous system findings					0.010 *
No	18 (60%)	77 (52.4%)	187 (50.3%)	108 (39.6%)	
Yes	12 (40%)	70 (47.6%)	185 (49.7%)	165 (60.4%)	
Pain in extremities	-	1 (1.43%)	2 (1.08%)	4 (2.42%)	
Vertigo	-	12 (17.14%)	50 (27.03%)	37 (22.42%)	
Numbness in fingers	-	4 (5.71%)	10 (5.41%)	6 (3.64%)	
Sweating	-	1 (1.43%)	1 (0.54%)	2 (1.21%)	
Febrile seizures	2 (16.67%)	5 (7.14%)	8 (4.32%)	18 (10.91%)	
Fainting	-	10 (14.29%)	23 (12.43%)	25 (15.15%)	
Epileptic seizures	8 (66.67%)	18 (25.71%)	32 (17.30%)	25 (15.15%)	
Nightmares		1 (1.43%)	3 (1.62%)	4 (2.42%)	
Developmental delay	4 (33.33%)	10 (14.29%)	17 (9.19%)	10 (6.06%)	
Frailty	-	-	2 (1.08%)	2 (1.21%)	
Headache	-	16 (22.86%)	36 (19.46%)	36 (21.82%)	
Sleep disorder	2 (16.67%)	3 (4.29%)	5 (2.70%)	-	

* *p* = 0.01.

## Data Availability

Not applicable.

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
