# Peer review of "Neurological Symptoms That May Represent a Warning in Terms of Diagnosis and Treatment in a Group of Children and Adolescents with Vitamin D Deficiency"

_children, 2023, doi:10.3390/children10071251_

Round 1

Reviewer 1 Report

The research problem presented by the authors relates to neurological symptoms occurring in children and adolescents diagnosed with vitamin D deficiency. Considering the important role of vitamin D in regulating calcium and phosphorus homeostasis and protecting bone health, this issue is important from the point of view of pediatric practice. The authors conducted a retrospective analysis of 822 cases of children and adolescents aged 1-18 with symptoms potentially associated with vitamin D deficiency and with low serum 25 OH vitamin D concentrations. The obtained results were subjected to statistical analysis. In the final conclusions, the authors stated that 'Movement disorders were present in 14.6% (n = 120) of the cases with vitamin D deficiency in this study (n = 822), and central nervous system findings in 52.6% (n = 432), and the incidences of these varied in terms of age groups, sex, and vitamin D levels'. It seems that these results are of significant importance for pediatric knowledge, therefore they should be of interest to paediatricians and other specialists.

From a formal point of view, the article is written correctly. Nevertheless, wanting to increase its value for a potential reader, I propose to change the title. It is not clear from the current title what kind of patients the research was conducted on. Therefore, it seems to me that the following title "Neurological symptoms in a group of children and adolescents with vitamin D deficiency" would be more adequate to the content of the article. In addition, in the 'Study design' section, criteria for inclusion or exclusion in the study should be added. Reading the text, I noticed a few grammatical mistakes, e.g., on p.1 (line 24) it is 'findig' instead of 'finding'. Therefore, I propose to carry out a linguistic correction of the text.

After making the proposed changes, I support the publication of the article.

Author Response

We are most grateful for your opinions concerning our study and your review thereof. We have up-dated the article accordingly. We would be most grateful if you would please review it once again.

Yours faithfully

The amendments made are as follows;

  1. The title has been changed
  2. More numerical data have been added to the Abstract
  3. The inclusion and exclusion criteria in the Methods section have been revised
  4. The fact that headache can represent an important warning has been emphasized in the Discussion section (42)
  5. Orthographic errors have been corrected

Reviewer 2 Report

The authors report data from a study about possible neurological presentations that can  represent warnings of Vitamin D deficiency in terms of diagnosis and treatmen. The topic is really interesting and the study is well organized and well written; however, I have few comments for the authors:

- Abstract: I suggest to add more numerical data about the results found.

- Methods: I suggest to describe better the methods used for the selection of patients: the authors should add the inclusion and exclusion criteria.

- Discussion: The authors must underline that headache can be an important warning (as found in their series): I think that the authors should read and cite the recent paper by Dell'Isola GB et al. J Clin Med. 2021 Dec 20;10(24):5983; the authors must remember that Vitamin D can useful for prevention of headache.

English language is good.

Author Response

(The authors gave the same response as above.)

Round 2

Reviewer 2 Report

No changes are requested.